# Two New Cytotoxic Steroidal Alkaloids from *Sarcococca Hookeriana*

**DOI:** 10.3390/molecules24010011

**Published:** 2018-12-20

**Authors:** Shaojie Huo, Jichun Wu, Xicheng He, Lutai Pan, Jiang Du

**Affiliations:** 1Guiyang College of Traditional Chinese Medicine, Guiyang 550025, China; 15038115764@163.com (S.H.); wujichun2018@sina.com (J.W.); hexicheng53@126.com (X.H.); 2Guizhou Provincial Key Laboratory of Miao Medicine, Guiyang 550025, China; ltpan@sina.cn

**Keywords:** Buxaceae, *Sarcococca hookeriana*, steroidal alkaloid, cytotoxicity

## Abstract

Two new steroidal alkaloids, named hookerianine A (**1**) and hookerianine B (**2**) were isolated from the stems and roots of *Sarcococca hookeriana* Baill., along with two known compounds, sarcorucinine G (**3**) and epipachysamine D (**4**). On the basis of spectroscopic methods and by comparison with literature data, their structures were determined. As well as X-ray crystallography was performed to confirm compound **4**. To identify novel antitumor inhibitors, all compounds were performed a CCK-8 assay against five human cancer cell lines SW480, SMMC-7721, PC3, MCF-7 and K562 in vitro. Compound **2** exhibited moderate cytotoxic activities to all cell lines with IC_50_ values in the range of 5.97–19.44 μM. Compound **3** was the most effective one against SW480 and K562 cell lines with IC_50_ values of 5.77 and 6.29 μM, respectively.

## 1. Introduction

The *Sarcococca* genus (Buxaceae) consists of about 20 species, widely distributed in the southwestern region of China and other south Asian countries [1]. The members of *Sarcococca* plants are used as TCM and traditional folk medicine for the treatment of stomach pain, rheumatism, swollen sore throat and traumatic injury [2,3,4]. Previous investigations on several species of this genus indicated that steroidal alkaloids are the major chemical components with a broad spectrum of biological activities, such as cholinesterase inhibition [5,6,7], antitumor [8], antibacterial [9], antileishamanial [10], antidiabetic [11] and estrogen biosynthesis-promoting [12].

*Sarcococca hookeriana*, one of *Sarcococca* plants, is usually confusedly used by ethnic minorities in China. Although dozens of steroidal alkaloids have been discovered from *S. hookeriana* of Nepal [13,14,15,16,17], there were few phytochemical or biological studies on this species which grows in China. Enlightened by the diverse bioactivities of steroidal alkaloids and the use of *Sarcococca* plants as folk medicine, *S. hookeriana* was chosen for searching antitumor agent by our research group and several cytotoxic steroidal alkaloids have been reported [18]. In continuation of our ongoing study on this plant, two new steroidal alkaloids, named hookerianine A (**1**) and hookerianine B (**2**), together with two known ones, sarcorucinine G (**3**) [19] and epipachysamine D (**4**) [20] (Figure 1), were characterized and their cytotoxicity were evaluated in vitro with a CCK-8 assay. Herein, we describe the isolation, structure elucidation and cytotoxicity of the isolates.

## 2. Results and Discussion

### 2.1. Elucidation of the Chemical Structure of Compounds

Hookerianine A (**1**) was obtained as white amorphous powder, positive to Dragendorff’s reagent. The molecular formula of C_31_H_48_N_2_O was determined by HR-ESI-MS (*m*/*z* 465.3833, [M + H]^+^). The IR absorption at 3294, 1637, 1539, 1496 and 760 cm^−1^ indicated the presence of a secondary amine, amide carbonyl and aromatic ring, respectively. The ^1^H-NMR spectra (Table 1) exhibited signals of five aromatic protons (*δ*_H_ 7.37, 7.33 and 7.28) and five methyls (*δ*_H_ 2.17, 0.87, 0.74 and 0.63). The ^13^C-NMR (Table 1) signals at (*δ*_C_ 135.5, 129.4, 129.1 and 127.4) were characteristic for a monosubstituted aromatic ring, whereas the signal at *δ*_C_ 170.1 was due to the carbonyl carbon. The NMR data of compound **1** was similar to epipachysamine D (**4**), having one more methylene. HMBCs (Figure 2) from H-C(2′) to C(1′), C(3′) and C(4′) indicated that the additional methylene was placed between C(1′) and C(3′). Thus, compound **1** possessed a novel phenylacetyl group instead of benzoyl group located at C(3). The relative configuration of C(3) was assigned as *α*-orientation by correlations of N-H with H*α*-C(1), H*α*-C(5), and H*β*-C(1) with H-C(19) in ROESY (Figure 2). Therefore, compound **1** was characterized as (20*S*)-20-(*N*,*N*-dimethylamino)-3*α*-phenylacetylamino -5*α*-pregnane, to which we give the trivial name hookerianine A.

Hookerianine B (**2**) was obtained as white amorphous powder also reacts positively with Dragendorff’s reagent. The molecular formula of C_30_H_44_N_2_O_2_ was determined by HR-ESI-MS (*m/z* 465.3471, [M + H]^+^). The IR absorption at 3402, 1644, 1603, 1521, 1488 and 718 cm^−1^ indicated the presence of a secondary amine, amide carbonyl and aromatic ring, respectively. The ^1^H-NMR spectra (Table 1) exhibited signals of five aromatic protons (*δ*_H_ 7.74, 7.47 and 7.41) and five methyls (*δ*_H_ 2.24, 1.14, 0.87 and 0.83). The ^13^C-NMR spectra (Table 1) displayed 30 carbon signals including one carbonyl carbon at (*δ*_C_ 166.7) and six carbons of an aromatic ring (*δ*_C_ 135.0, 131.2, 128.5 and 126.8), respectively. whereas the signals at (*δ*_C_ 73.6 and 60.9) were due to two oxygenated carbons. The NMR data of compound **2** was similar to epipachysamine D (**4**), and the difference was the downfield chemical shift of C(16) and C(17) at *δ*(C) 73.6 and 60.9, which suggested that compound **2** possessed an epoxy group at C(16) and C(17), confirmed by HMBCs (Figure 3) from H-C(16) to C(14) and C(15), from H-C(18) and H-C(21) to C(15). The ROESY correlations (Figure 3) of N-H with H*α*-C(5), and H*β*-C(16) with N-Me suggested that the substituent at C(3) and the epoxy group at C(16) and C(17) all had *α*-orientations. Thus, compound **2** was characterized as (20*S*)-20-(*N*,*N*-dimethylamino)-16*α*,17*α*-epoxy-3*α*-benzoylamino-5*α*-pregnane, to which we give the trivial name hookerianine B.

The structures of known compounds **3**–**4** were determined by comparing their spectral data with literature data. To further confirm the chemical structure of compound **4**, a colorless crystal was obtained from CH_2_Cl_2_, and X-ray crystallography analysis with Mo *K*α radiation was performed. Through structural refinement by direct method SHELX-2014 [21,22], the chemical structure of **4** was identified as shown in Figure 4.

### 2.2. Results of the Cytotoxicity Test

The IC_50_ values of four compounds against five human cancer cell lines: SW480, SMMC-7721, PC3, MCF-7 and K562 are summarized in Table 2 (DDP and 5-FU was used as the positive control). The compound **2**, a new steroidal alkaloid, exhibited moderate cytotoxic activities to all cell lines with IC_50_ values in the range of 5.97–19.44 μM. Compared to the positive control 5-FU with IC_50_ values of 7.65 and 4.78 μM against SW480 and K562 cell lines, the compound **3** was the most effective one against these cell lines with IC_50_ values of 5.77 and 6.29 μM, respectively. The structure-activity relationships of compound **1** and **4** showed that steroidal alkaloids possessed a novel phenylacetyl group instead of benzoyl group located at C-3 can increase the cytotoxicity to human cancer cell lines: SW480, SMMC-7721, PC3 and K562. Interestingly, the cytotoxicity of compound **3** is stronger than compound **4**, which indicated that the presence of double bond between C-16 and C-17 can increase the cytotoxicity. Meanwhile, compared to compound **4**, compound **2** possessed an epoxy group at C-16 and C-17 also showed better cytotoxicity. The results suggested that C-16 and C-17 of steroidal alkaloids play an important role in anticancer potential.

## 3. Materials and Methods

### 3.1. General Experimental Procedures

Optical rotations were measured with a Rudolph Autopol I automatic polarimeter (Rudolph, Hackettstown, NJ, USA). UV spectra were obtained on a Shimadzu UV-2401PC spectrophotometer (Shimadzu, Kyoto, Japan). IR spectra were measured with a Bruker TENSOR-27 spectrophotometer (Bruker, Bremerhaven, Germany) using KBr pellets. The 1D and 2D NMR spectra were recorded on JEOL ECX 500 MHz spectrometers (JEOL Ltd, Kyoto, Japan) with TMS as an internal standard. Chemical shifts (*δ*) were expressed in ppm with reference to solvent signals. High-Resolution Electrospray Ionization Mass Spectrometry (HR-ESI-MS) was recorded on a Bruker Daltonics micrOTOF-Q II spectrometer (Bruker, Bremerhaven, Germany). Column chromatography (CC) was performed on Silica gel (200–300 and 300–400 mesh, Qingdao Marine Chemical Ltd., Qingdao, China). Fractions were monitored by TLC (GF 254, Qingdao Haiyang Chemical Co., Ltd., Qingdao, China), and spots were visualized by Dragendorff’s reagent. Solvents were distilled prior to use for extraction and isolation.

### 3.2. Plant Material

The plants of *S. hookeriana* were collected from Hezhang Country, Guizhou Province of China, in July 2015 and identified by Prof. JunHua Zhao*,* Guiyang College of Traditional Chinese Medicine. A voucher specimen (No. 150708) was deposited at College of Pharmacy, Guiyang College of Traditional Chinese Medicine.

### 3.3. Extraction and Isolation

The powdered stems and roots of *S. hookeriana* (14.5 Kg) were extracted ultrasonically with MeOH for three times. The combined extracts were concentrated and then partitioned between EtOAc and 1% aq. H_2_SO_4_. The acid-soluble fraction was alkalinized with aq. Na_2_CO_3_ to pH 9 and followed by exhaustive extraction with CH_2_Cl_2_ to afford crude alkaloids (156 g). The crude alkaloids were roughly separated by CC (SiO_2_; CH_2_Cl_2_/MeOH/Et_2_NH, 100:0:0→10:1:0→5:1:1) to give five fractions: *Frs. A-E*. *Fr. A* (32 g) was passed through CC [SiO_2_; petroleum ether (PE)/CH_2_Cl_2_/Et_2_NH 50:1:1→10:1:1, then cyclohexane/acetone/Et_2_NH 20:1:1] to afford **3** (150 mg), **4** (800 mg). *Fr. B* (24 g) was subjected to CC (PE/CH_2_Cl_2_/ Et_2_NH, 50:1:1→20:1:1, then CH_2_Cl_2_/MeOH 20:1) to yield **1** (60 mg) and **2** (40 mg).

#### 3.3.1. Compound **1**

The Hookerianine A (**1**): White amorphous powder. ${\left[\alpha \right]}_{D}^{14}$ = +21.2 (*c* = 0.565, CH_2_Cl_2_). UV (CHCl_3_) λ_max_ (log *ε*) 242.0 (0.25) nm. IR (KBr) υ_max_: 3294, 3029, 2929, 2865, 2761, 1637, 1539, 1496, 760 cm^−1^. ^1^H- and ^13^C-NMR data are shown in Table 1. HR-ESI-MS *m*/*z* 465.3833 ([M + H]^+^, C_31_H_49_N_2_O^+^; calc. 465.3839).

#### 3.3.2. Compound **2**

Hookerianine B (**2**): White amorphous powder. ${\left[\alpha \right]}_{D}^{14}$ = +21.2 = +3.7 (c = 0.092, CH_2_Cl_2_). UV (CHCl_3_) λ_max_ (log *ε*) 244.5 (1.53). IR (KBr) υ_max_: 3402, 3032, 2930, 2853, 2767, 1644, 1603, 1521, 1488, 718, 694 cm^−1^. ^1^H- and ^13^C-NMR data are shown in Table 1. HR-ESI-MS *m*/*z* 465.3471 ([M + H] ^+^, C_30_H_45_N_2_O_2_^+^; calc. 465.3476).

### 3.4. Single Crystal X-Ray Data of Compound ***4***

Crystal data of **4** (from CH_2_Cl_2_): C_30_H_46_N_2_O, M = 450.69, space group *P*2_1_ (No. 4), monoclinic, *Z* = 2, *a* = 5.895(14) Å, *b* = 9.983(2) Å, *c* = 22.033(5) Å, *α* = 90°, *β* = 95.971(6)°, *γ* = 90°, *V* = 1289.4(5)Å^3^, T = 173 K, μ (Mo *K*α) = 0.71073 mm^−1^. A crystal of dimensions of 0.18 × 0.08 × 0.05 mm^3^ was measured on a Bruker APEX-II CCD diffractometer with a graphite monochromator (*φ*-*ω* scans, 2θ_max_ = 55.18°), Mo *K*α radiation. 9787 reflections were measured, 5785 independent reflections were observed (R_int_ = 0.0530). The final R_1_ values were 0.0625 (I >= 2σ (I)). The final wR_2_ values were 0.1278 (I >= 2σ (I)). The final R_1_ values were 0.1008 (all data). The final wR_2_ values were 0.1449 (all data). The goodness of fit on *F*^2^ was 0.971. CCDC 1875789 for compound **4** contains the supplementary crystallographic data for this paper. These data can be obtained free of charge via https://www.ccdc.cam.ac.uk/.

### 3.5. Cytotoxicity Assay

To identify novel antitumor inhibitors, compounds **1**–**4** were tested on five human cancer cell lines SW480, SMMC-7721, PC3, MCF-7 and K562 by using a CCK-8 assay. All cells were obtained from Centre of Drug Safety Evaluation and Research of Hunan Province. Those cells were cultured in a DMEM medium (high glucose) (Hyclone, Logan, UT, USA), which was supplemented with 10% fetal bovine serum (Sciencell, San Diego, CA, USA) in a humidified 5% CO_2_ atmosphere at 37 °C. CCK-8 was purchased from American Bimake Company (Bimake, Houston, TE, USA).

The cytotoxicity assay was performed according to the Cell Counting Kit-8 assay methods as described by elsewhere [23]. Briefly, all cells were seeded into 96-well plates at 3 × 10^3^ cells per well and allowed to culture for 12 h before the addition of the drug. Then, each tumor cell line was exposed to the tested compounds at different concentrations (100–0 μM) for 72 h. DDP (Tokyo Chemical Industry, Tokyo, Japan) and 5-FU (Amresco, Portland, ME, USA) was used as positive control. After treatment, 10 μL of CCK-8 was added to each well, and the plates were incubated for an additional 12 h. OD_450_ absorbance was determined using a Spectramax-i3x (Molecular Devices, Sunnyvale, CA, USA). The experiments were performed in triplicate to obtained IC_50_ values.

## 4. Conclusions

In this study, two new steroidal alkaloids, hookerianine A (**1**) and hookerianine B (**2**), together with two known ones, scorucinine G (**3**) and epipachysamine D (**4**), were isolated from the stems and roots of *S. hookeriana*. To the best of our knowledge, four compounds were isolated from this plant for the first time. Two new compounds were shown to possess a 3*α* substituent, which were rarely reported. In addition, compound **1** represents the first example of pregnane-type steroidal alkaloid possessed a novel phenylacetyl group at C-3. Based on the preliminary structure-activity relationships study, we found that the different substituents at C-3 and the presence of double bond and epoxy group between C-16 and C-17 have an important effect on the cytotoxicity of steroidal alkaloids. Inspired by this, it deserves further structural modification and in-depth mechanism research on steroid alkaloids with those characteristics. The results suggested that these types of steroidal alkaloids may have the potential to be anticancer agents.

All of the ^1^H-NMR, ^13^C-NMR, 2D-NMR and HR-ESI-MS spectra of compound **1** and **2** are available in Appendix A.

## Figures and Tables

**Figure 1 molecules-24-00011-f001:**
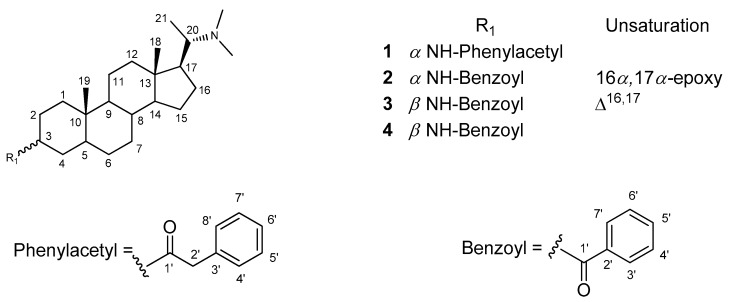
Structures of compounds **1**–**4**.

**Figure 2 molecules-24-00011-f002:**
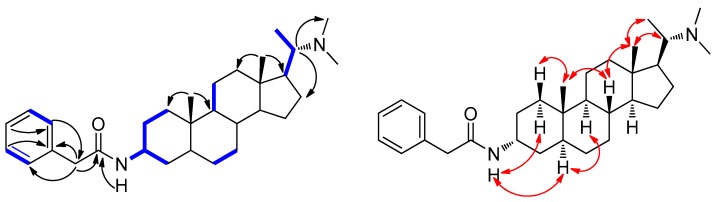
Key ^1^H,^1^H-COSY (
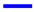
), HMBC (H→C), and ROESY (
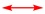
) correlations of compound **1**.

**Figure 3 molecules-24-00011-f003:**
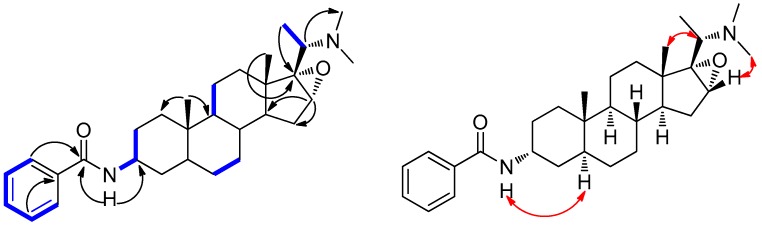
Key ^1^H,^1^H-COSY (
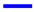
), HMBC (H→C), and ROESY (
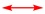
) correlations of compound **2**.

**Figure 4 molecules-24-00011-f004:**
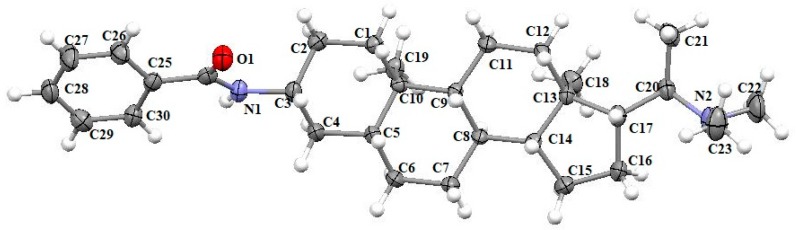
ORTEP drawing of compound **4.**

**Table 1 molecules-24-00011-t001:** ^1^H- (500 MHz) and ^13^C- (125 MHz) NMR data of compounds **1**–**2** in CDCl_3_.

Position	1		2	
	*δ*_H_ (*J* in Hz)	*δ* _C_	*δ*_H_ (*J* in Hz)	*δ* _C_
1	1.45 (m), 0.65 (m)	33.2 (t)	1.72 (m), 1.10 (m)	37.2 (t)
2	1.63 (m), 1.52 (m)	25.9 (t)	1.93 (d, *J* = 14.8), 1.38 (m)	28.8 (t)
3	4.06 (m)	44.8 (d)	3.96 (m)	49.3 (d)
4	1.45 (m), 1.28 (m)	32.7 (t)	1.70 (m), 1.24 (m)	35.4 (t)
5	0.79 (m)	41.0 (d)	1.24 (m)	45.4 (d)
6	1.12 (m)	28.5 (t)	1.32 (m), 1.24 (m)	28.4 (t)
7	1.64 (m), 0.77 (m)	32.1 (t)	1.62 (m), 0.95 (m)	31.6 (t)
8	1.31 (m)	35.4 (d)	1.62 (m)	33.8 (d)
9	0.46 (d, *J* = 11.8, 3.9)	54.7 (d)	0.73 (m)	54.5 (d)
10		36.0 (s)		35.6 (s)
11	1.42 (m), 1.18 (m)	20.8 (t)	1.62 (m), 1.29 (m)	20.8 (t)
12	1.87 (m), 1.08 (m)	39.8 (t)	1.62 (m), 1.47 (m)	32.8 (t)
13		41.7 (s)		42.2 (s)
14	1.01 (m)	56.8 (d)	1.24 (m)	45.1 (d)
15	1.58 (m), 1.02 (m)	24.0 (t)	1.84 (dd, *J* = 12.0, 4.8), 1.20 (m)	27.2 (t)
16	1.85 (m), 1.44 (m)	27.7 (t)	3.55 (s)	60.9 (d)
17	1.36 (d, *J* = 9.9)	54.9 (d)		73.6 (s)
18	0.63 (s)	12.4 (q)	0.87 (s)	15.8 (q)
19	0.74 (s)	11.5 (q)	0.83 (s)	12.2 (q)
20	2.41 (dq, *J* = 10.2, 6.4)	61.2 (d)	2.84 (q, *J* = 6.6)	55.7 (d)
21	0.87 (s)	10.0 (q)	1.14 (s)	13.6 (q)
N(Me)2	2.17 (s)	39.9 (q)	2.24 (s)	43.1 (q)
1′		170.1 (s)		166.7 (s)
2′	3.57 (dt, *J* = 6.8, 1.2)	44.2 (t)		135.0 (s)
3′		135.5 (s)	7.74 (dt, *J* = 6.8, 1.2)	126.8 (d)
4′	7.28 (m)	129.4 (d)	7.41 (m)	128.5 (d)
5	7.37 (m)	129.1 (d)	7.47 (m)	131.2 (d)
6′	7.33 (m)	127.4 (d)	7.41 (m)	128.5 (d)
7′	7.37 (m)	129.1 (d)	7.47 (m)	126.8 (d)
8′	7.28 (m)	129.4 (d)		
NH	5.66 (d, *J* = 8.0)		5.98 (d, *J* = 8.0)	

**Table 2 molecules-24-00011-t002:** Cytotoxicity of compounds **1**–**4** against SW480, SMMC-7721, PC3, MCF-7 and K562 cells in vitro.

Compounds	IC_50_ (μM) ^a^ (*n* = 3)				
	SW480	SMMC-7721	PC3	MCF-7	K562
**1**	10.97 ± 1.36	41.31 ± 3.02	32.97 ± 3.78	37.30 ± 0.99	11.86 ± 0.82
**2**	5.97 ± 0.13	16.19 ± 0.56	11.57 ± 0.86	19.44 ± 1.70	7.95 ± 0.02
**3**	5.77 ± 0.29	10.84 ± 1.19	11.79 ± 2.96	44.97 ± 4.73	6.29 ± 0.53
**4**	45.92 ± 1.56	71.13 ± 5.37	>100	28.92 ± 1.22	85.48 ± 6.77
DDP ^b^	4.71 ± 0.20	4.03 ± 0.62	6.50 ± 0.44	6.86 ± 0.42	5.49 ± 0.83
5-FU ^c^	7.65 ± 0.26	7.86 ± 0.38	8.18 ± 0.73	6.74 ± 0.89	4.78 ± 0.27

^a^ Values of IC_50_ expressed as mean ± SD, *n* = 3 for all groups. ^b^ DDP, the abbreviation of cisplatin, used as reference drug. ^c^ 5-FU, the abbreviation of 5-fluorouracil, used as reference drug.

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
