# Peer review of "Two New Cytotoxic Steroidal Alkaloids from Sarcococca Hookeriana"

_molecules, 2018, doi:10.3390/molecules24010011_

Round 1

Reviewer 1 Report

The manuscript entitled: "Two new cytotoxic steroidal alkaloids from Sarcococca hookeriana" represents an interesting contribution to evaluate natural compounds for cancer treating. However, the authors should improve their paper proposing at least some mechanisms of action of these molecules. In addition, they should explain why these molecules are not effective on different tumoral cell lines where the reference compounds are always effective!

Author Response

Dear Prof. Letty Zhu,

Thank you very much for your timely informing us the remarks on our manuscript entitled “Two new cytotoxic steroidal alkaloids from Sarcococca hookeriana (molecules-409242)”. Based on the reviewers′ comments, we have checked and revised the manuscript carefully. The reply to the reviewer’s suggestion and the detailed modification of our paper were given as follows.

Point 1: However, the authors should improve their paper proposing at least some mechanisms of action of these molecules.

Thanks for your advice. Isolation of compound and investigation on their bioactivity is part of our research. We commit to find out the anticancer drugs applied to clinic from steroidal alkaloids, so there is a long way to go. Based on the preliminary structure-activity relationships study, we found that the different substituents at C-3 and the presence of double bond and epoxy group between C-16 and C-17 have an important effect on the cytotoxicity of steroidal alkaloids. Inspired by this, we are conducting structural modification and in-depth mechanism research on steroid alkaloids with those characteristics. Of course, once we have the results of the research, we will publish soon.

Point 2: In addition, they should explain why these molecules are not effective on different tumor cell lines where the reference compounds are always effective!

All experiments were carried out strictly in accordance with the rules of CCK8. The reference drugs we used, such as DDP and 5-Fu, generally have no selective inhibition activity against all cells. However, some tested alkaloids present very different or no cytotoxicity to different cell lines. We also refer to articles about the cytotoxic activity of steroid alkaloids, such as “Zhang, P.; Shao, L.; Shi, Z. Pregnane alkaloids from Sarcococca ruscifolia and their cytotoxic activity and Yan, Y.X.; Sun, Y.; Chen, J.C. Cytotoxic steroids from Sarcococca saligna” which have the similar result. This indicates that steroid alkaloids have selective inhibition activity against cancer cell lines and may be harmless to some cancer cell lines, which is beneficial for our subsequent studies. Steroid alkaloids have selective inhibition activity against different tumor cell lines may affected by the different mechanism and we will further study that. Based on your comments, we have checked and revised the manuscript carefully. Thank you very much for your good suggestion which is helpful for our future research.

We are very gratitude to you and the reviewer’s patient for checking our manuscript, which is very useful for us to organize our paper. We hope the revised manuscript could meet your requirement. Again, thank you very much for your efforts.

Reviewer 2 Report

    The manuscript entitled"Two new cytotoxic steroidal alkaloids from Sarcococca hookeriana" is a well written article with good explanation of the experimental methods used and all data collected. It has significant work done as they first time isolated those four compound. Isolation procedure and the description was fine. The point is missing here is an  invivo experiment. They have shown IC50/cytotoxicity in five cell lines and has shown the effects of those compound. Now they need to check the efficacy of those drug into a diseased animal

Author Response

Dear Prof. Letty Zhu,

Thank you very much for your timely informing us the remarks on our manuscript entitled “Two new cytotoxic steroidal alkaloids from Sarcococca hookeriana (molecules-409242)”. Based on the reviewers′ comments, we have checked and revised the manuscript carefully. The reply to the reviewer’s suggestion and the detailed modification of our paper were given as follows.

Point 1:The point is missing here is an invivo experiment. They have shown IC50/cytotoxicity in five cell lines and have shown the effects of those compounds. Now they need to check the efficacy of those drugs into a diseased animal.

Based on the preliminary structure-activity relationship study, we found that the different substituents at C-3 and the presence of double bond and epoxy group between C-16 and C-17 have an important effect on the cytotoxicity of steroidal alkaloids. Inspired by this, we are conducting structural modification to increase the solubility and pharmacodynamics of the steroid alkaloids for further in vivo experiment and in-depth mechanism research. We are working on this and have made some progress so far. This paper refers to two new compounds and as researchers we wish our outcome published soon. According to your comments, we have checked and revised the manuscript carefully. Thank you very much for your good suggestion.

We are very gratitude to you and the reviewer’s patient for checking our manuscript, which is very useful for us to organize our paper. We hope the revised manuscript could meet your requirement. Again, thank you very much for your efforts.

Round 2

Reviewer 1 Report

The manuscript may be accepted in the present form.